# A Novel Algorithm for Evaluating Bone Metastatic Potential of Breast Cancer through Morphometry and Computational Mathematics

**DOI:** 10.3390/diagnostics13213338

**Published:** 2023-10-30

**Authors:** Simona-Alina Duca-Barbu, Alexandru Adrian Bratei, Antonia-Carmen Lisievici, Tiberiu Augustin Georgescu, Bianca Mihaela Nemes, Maria Sajin, Florinel Pop

**Affiliations:** 1Department of Pathology, “Carol Davila” Clinical Nephrology Hospital, 010731 Bucharest, Romania; 2Department of Pathology, University of Medicine and Pharmacy “Carol Davila”, 020022 Bucharest, Romania; 3Faculty of Chemical Engineering and Biotechnologies, University Politehnica of Bucharest, 011061 Bucharest, Romania; 4Laboratory of Electrochemistry and PATLAB, National Institute of Research for Electrochemistry and Condensed Matter, 060021 Bucharest, Romania; 5Department of Pathology, National Institute for Mother and Child Health, 011061 Bucharest, Romania; 6International Computer High School of Bucharest, 032622 Bucharest, Romania

**Keywords:** breast cancer, bone metastasis, tumor morphometry, digital pathology

## Abstract

Bone metastases represent about 70% of breast cancer metastases and are associated with worse prognosis as the tumor cells acquire more aggressive features. The selection and investigation of patients with a high risk of developing bone metastasis would have a significant impact on patients’ management and survival. The patients were selected from the database of Carol Davila Clinical Nephrology Hospital of Bucharest. Their tumor specimens were pathologically processed, and a representative area was selected. This area was scanned using an Olympus VS200 slide scanner and further analyzed using QuPath software v0.4.4. A representative group of approximately 60–100 tumor cells was selected from each section, for which the following parameters were analyzed: nuclear area, nuclear perimeter, long axis and cell surface. Starting from these measurements, the following were calculated: the mean nuclear area and mean nuclear volume, the nucleus to cytoplasm ratio, the length of the two axes, the long axis to short axis ratio, the acyclicity and anellipticity grade and the mean internuclear distance. The tumor cells belonging to patients known to have bone metastasis seemed to have a lower nuclear area (<55 µm^2^, *p* = 0.0035), smaller long axis (<9 µm, *p* = 0.0015), smaller values for the small axis (<7 µm, *p* = 0.0008), smaller mean nuclear volume (<200 µm^3^, *p* = 0.0146) and lower mean internuclear distance (<10.5 µm, *p* = 0.0007) but a higher nucleus to cytoplasm ratio (>1.1, *p* = 0.0418), higher axis ratio (>1.2, *p* = 0.088), higher acyclicity grade (>1.145, *p* = 0.0857) and higher anellipticity grade (>1.14, *p* = 0.1362). These parameters can be used for the evaluation of risk category of developing bone metastases. These results can be useful for the evaluation of bone metastatic potential of breast cancer and for the selection of high-risk patients whose molecular profiles would require further investigations and evaluation.

## 1. Introduction

Breast cancer has become one of the most common neoplasms in females and is also the most commonly diagnosed cancer type, with its burden growing in many parts of the world. Breast cancer occurs in women at any age, with increasing rates in later life, but may also occur in the male population [1,2,3,4]. Being the most prevalent malignancy, it is also the most common cause of death related to neoplastic disease in women, due to the development of distant metastases, with bone being the most frequent site of dissemination [5,6,7].

Breast cancer is a complex disease with various subtypes, featuring distinct histopathological characteristics and immunohistochemical profiles and specific clinical behavior and outcome [1,8,9]. Breast cancers are classified according to the histological and molecular features of the tumor [1]. One of the most common malignant lesions, accounting for more than 95% of all breast cancers, are breast carcinomas, adenocarcinomas in particular [1,10].

Histologically, the predominant subtypes of invasive breast carcinoma include invasive ductal carcinoma (typically exhibiting positive membrane E-cadherin expression) and invasive lobular carcinoma (typically exhibiting negative membrane E-cadherin expression), with the former being the most prevalent [1,10,11]. In addition to these primary types, invasive breast carcinoma encompasses various less common subtypes, such as mucinous, micropapillary, metaplastic, tubular and cribriform, among others [12].

In the case of invasive ductal carcinoma (IDC), it is further categorized into various histological special subtypes based on distinctive cytoarchitectural features (papillary, medullary, etc.). However, most cases lack sufficient morphological characteristics to be classified as a distinct subtype and are denoted as IDC with no special type or NST [1,3,13,14].

Microscopically, it is characterized by a heterogenous growth pattern: diffuse, corded, nested or single cells with variable ductal differentiation. The neoplastic cells may have various shapes and sizes, with pleomorphic nuclei, prominent nucleoli and frequent mitoses. Areas of necrosis, calcification, squamous and apocrine metaplasia can be present, while the stromal tissue can show different degrees of desmoplasia [1,15].

Invasive lobular carcinoma (ILC) is the second most common histological type of invasive breast carcinoma, with distinct biological features and outcome. It usually affects women who are older than the ones affected by other types of breast cancer [1,3]. The histopathological features of ILC include small, round, noncohesive tumor cells with single-file growth patterns. The most characteristic molecular change is the inactivation of E-cadherin, a useful immunohistochemical marker used in the diagnosis of ILC [15,16,17].

In the past decades, numerous studies were conducted to understand the underlying molecular features of breast carcinomas. Perou et al. were among the first authors to discover and describe the heterogeneity of breast carcinomas by classifying them into distinct subgroups, based on their molecular features, using microarray technology [1,5,12]. The molecular subtypes of breast cancer, as currently defined by the surrogate classification system, are as follows:Luminal A: Estrogen (ER)/Progesteron (PR) positive or HER2 negative. It represents about 50% of invasive breast cancer and has a good prognosis [15];Luminal B: ER/PR positive with HER2/neu positive or negative. It is the second most common type, after Luminal A, but the response to endocrine therapy is variable, which means that the prognosis is poorer [15];HER2 overexpression: ER/PR negative with HER2/neu strongly positive. It is frequently associated with high Ki-67 and TP53 mutation and has a poor prognosis [15];Basal-like: CK5/6 and/or EGFR positive, ER/PR and HER2 negative with high Ki-67 and TP53 mutation. It has no response to endocrine therapy, and it generally has a poor prognosis [15].

These subtypes have different prognoses, metastatic patterns, and therapeutic approaches [18,19]. Metastasis is a complex process involving multiple mechanisms: division of tumor cells from the primary tumor, invasion, migration, extravasation and regulation of the microenvironment [18,20,21,22]. The most common sites of distant metastasis include the liver, lungs, brain and bone, the latter being the most frequent location involved in more than 70% of patients with metastatic breast carcinomas [5,23,24,25]. Metastatic tumors frequently develop primarily in the axial skeleton: spine, ribs, proximal femur [21,26,27,28].

Recent studies based on the metastatic potential of different subtypes of breast cancer have shown that although HER-2-positive and triple-negative cancers are more aggressive and more likely to develop metastases in general, patients with ER/PR positive tumors tend to show a higher incidence of bone metastasis, specifically [5,27,29,30,31,32].

Metastatic breast cancer cells acquire aggressive characteristics leading to therapy failure and the death of many patients. Because patient prognosis is closely related to metastasis development, it is important to predict distant spread in the early stages because it can increase survival rates [18,21].

Bone metastases are a serious complication of breast cancer that can affect the quality of life and survival of patients by causing pain, fractures, spinal cord compression, nerve damage and high levels of calcium in the blood, which can lead to nausea, vomiting, confusion and kidney problems.

Identifying high-risk patients for bone metastases is important for several reasons. First, it can help prevent or delay the onset of bone complications by initiating early treatment with drugs that can strengthen the bones and reduce the risk of fractures. Second, it can help monitor the response to treatment and adjust the therapy accordingly. Third, it can help improve the prognosis and quality of life of patients by reducing the symptoms and complications of bone metastases.

Assessing the risk of bone metastases in breast cancer patients allows the clinician to tailor their treatment accordingly. For example, patients with a high risk of bone metastases may benefit from receiving bisphosphonates or denosumab, which are drugs that can prevent or slow down bone loss and reduce the risk of fractures. Patients with a low risk of bone metastases may not need these drugs or may receive them at a lower dose or frequency.

Given the importance of this pathology in clinical practice, in this article we aim to evaluate the morphometric characteristics of tumor cells in an attempt to find statistically significant differences between the ones that are associated with bone metastases and the ones that are not.

## 2. Materials and Methods

### 2.1. Patients’ Description

A total of 41 patients diagnosed with invasive breast carcinoma of no special type (NST) between 2014 and 2022 were selected from the internal database of Carol Davila Clinical Hospital. Among them, a total of 7 patients were histopathologically confirmed to have bone metastases, while the other 34 had not developed any clinical signs of bone metastases before or after surgical treatment.

### 2.2. Apparatus

The histopathological slides containing tumor sections were scanned using Olympus VS200 slide scanner at high focusing and 40× objective. After obtaining the virtual version of the slides, the files were transferred to another computer, where measurements were performed using Qupath open-source software v0.4.4. [33].

The privacy of the patients was preserved during our research, as all the patients were identified by numbers, their slide number was anonymized, and all the virtual slides were saved on an external hard disk belonging to the pathology lab of the hospital.

### 2.3. Methods

#### 2.3.1. Selection of the Representative Slide and Tumor Area

For each virtual slide, a representative area was selected. The criteria for selecting this area were represented by tumor cells with characteristic morphologic features, high mitotic index, high cell density (nests or cell islands), less or no empty spaces (like vacuoles, obvious discohesive areas), lack of necrosis or abundant inflammatory infiltrate and, most importantly, large and non-overlapping nuclei.

After selecting these areas, 60–100 cells per representative area, whose nuclei were larger and non-overposing, were chosen. The reasoning for selecting cells with larger nuclei was to avoid the risk of erroneous measurement interpretation due to the lack of coplanarity, as smaller-sized nuclei may represent sections of larger nuclei located in the upper or lower planes and therefore not reflect the real dimensions of the cells. The necessity for nuclei to be non-overlapping is to ensure a correct measurement of nuclear dimensions. As criterion for excluding smaller nuclei, there were selected only the nuclei with an area bigger than half the average dimension of the largest non-overlapping nuclei.

#### 2.3.2. Obtaining the Morphometrical Data

For this paper, the aim was to gather data about the morphometric features of tumor cells. To achieve this, the following were evaluated: mean area of nuclei, mean nuclear volume, the two axes, the ratio between long and small axes, the surface ratio of nucleus to cytoplasm, the acyclicity and anellipticity grades and mean internuclear distance as described below.

In order to obtain the above-described parameters, for each selected cell the nuclear area, nuclear perimeter and long axis of nucleus were measured (Figure 1).

For nucleus to cytoplasm ratio and for mean distance between cells, the cytoplasm had to be quantified. In cells where the cellular border could be evaluated, the perimeter and area of the whole cell were measured (Figure 1 (right) and Figure 2 (left)). If the cellular margins could be evaluated only partially, small areas including more cells were selected and the cytoplasm was considered to be uniformly distributed to each cell, and, for each cell, the nucleus was considered to have the mean average area (Figure 2 (right)).

#### 2.3.3. Calculation of the Morphometrical Parameters

The average nuclear area was calculated from the nuclear area of each cell.

For evaluation of axes ratio, the small axis was calculated in the hypothesis of an elliptic nuclei, whose area is given by Equation (1):(1)$$A=\pi \cdot r_{1}\cdot r_{2}$$
where *A* is the nuclear area of each cell, *r*_1_ is half of the long axis and *r*_2_ is half the small axis. After calculating the small axis length, the ratio could be evaluated.

The evaluation of the nucleus to cytoplasm surface ratio, there were taken into consideration both the cases discussed above. For the cells with evaluable cellular membranes, the surface belonging to cytoplasm was calculated as the difference between cell surface and nuclear surface, and the ratio was calculated for each cell and averaged for each patient. On the other hand, for only partially evaluable cellular membranes, all the selected areas of the counted nuclei were summed up. The area belonging to cytoplasm was calculated as the difference between the whole cellular surface, and the area belonging to the counted nuclei was considered to have the mean average surface. The cytoplasm corresponding to each cell was calculated assuming that each cell has the same amount of cytoplasm. Further, the mean ratio nucleus to cytoplasm could be calculated.

The acyclicity grade evaluates the deviation of the nucleus from the classical round form during the process of turning to malignant cells and was calculated as the ratio between the cell perimeter and the perimeter of a circle with the same area.

The next parameter taken into consideration in this paper is represented by the mean internuclear distance. This distance is measured in the high cellular areas where the stroma is minimal, as the quantity of stroma may influence the distance. As the nuclei were not equidistant, an assumption had to be made and, therefore, the principles of thermodynamics were applied as follows: in a vessel containing fluid in which the volume and the number of molecules are known, the mean distance is calculated by assuming that each molecule is the center of a cube. Therefore, the number of cubes is equal to the number of molecules, and the volume of each cube can be obtained by dividing the volume to the number of cubes. As each molecule is considered the center of a cube, the distance from each molecule center to one of the cube faces is equal to the half of the edge of the cube. Moreover, each cube would share a face with each adjacent cube. This aspect is important as each cube face is located at the half point on the distance between two molecules. Therefore, the mean intermolecular distance would be twice the distance from a molecule to the center of a cube face and as this distance is half the edge of the cube, the mean intermolecular distance would be equal to the edge of the cube. The adaptation to a two-dimensional plane would be made by dividing the area into equal squares. By following a similar algorithm, the mean distance would be equal to the side of a square. In morphometry, the whole measured cellular area has to be divided into squares, and each nucleus is considered the center of such a square. By applying the algorithm for planes as the morphometric measurements are carried out on a section, the square area would be equal to the mean cell area and the mean internuclear distance would be equal to the root square of the mean cell area.

Other nuclear measures are reflected in the length of the two axes and the approximation of nuclear volume. In order to approximate the volume of a nucleus, another assumption has to be made. The nucleus is considered an ellipsoid, in which two axes are considered equal to the small axis, and the third axis is equal to the long axis of the ellipse in plane. The volume of an ellipsoid adapted to this assumption can be calculated with the next formulae, given in Equation (2):(2)$$V=\frac{4}{3}\cdot \pi \cdot r_{1}\cdot r_{2}^{2}$$

The last evaluated parameter was the anellipticity grade. It is defined as the ratio between the nuclear perimeter and the perimeter of an ellipse with the long axis and small axis of the nucleus. The formulae for approximation of ellipse perimeter adapted to the assumption is given by Equation (3):(3)$$P=2\cdot \pi \cdot \sqrt{\frac{r_{1}^{2}+r_{2}^{2}}{2}}$$

The more folded the nucleus is, the higher the acyclicity and anellipticity grades are.

## 3. Results

All the above presented parameters were evaluated in both the tumors with associated bone metastases and those without. The results are given in Table 1.

Analyzing the data shown in Table 1, we tried to establish statistically significant cut-offs between the group of patients with bone metastases and the group without.

Therefore, it was observed that all tumors with associated bone metastases have a nuclear area < 55 µm^2^, while 71.42% of the patients without bone metastases have a nuclear area > 55 µm^2^.

In regard to the N/C ratio, 85.71% of the patients with tumors associating bone metastases have a N/C ratio > 1.1, while 71.42% of the patients without bone metastases have a N/C ratio < 1.1.

On the other hand, all the patients with breast cancers with associated bone metastases have an axis ratio > 1.2, and most of them have the value of this ratio over 1.26.

Regarding the acyclicity grade, 85.71% of the patients with tumors associated with bone metastases have an acyclicity grade > 1.145, and 60.71% of the patients without bone metastases have values of acyclicity grade < 1.145.

The mean internuclear distance for all the patients with tumors with associated bone metastases is observed to be <9.5 µm, while for patients without bone metastases, 82.14% have mean distances >9.5 µm.

In a similar manner, for the long axis, the chosen cut-off value was 9 µm, as 85.71% of the patients with tumors with associated bone metastases have a long axis length < 9 µm, and 82.35% of the patients without bone metastases have a long axis length > 9 µm. For the small axis, the chosen cut-off value was 7 µm, as all the patients with tumors with associated bone metastases have a small axis length < 7 µm, and 91.18% of the patients without known bone metastases have a small axis length > 7 µm.

The nuclear volume, as defined for the assumption of an ellipsoid, was given the cut-off value of 200 µm^3^, as 71.43% of the patients with tumors with associated bone metastases have the mean nuclear volume < 200 µm^3^, and 97.06% of the patients without known bone metastases have the long axis length > 200 µm^3^.

The last evaluated parameter is the anellipticity grade, as 85.71% of the patients with tumors with associated bone metastases have the anellipticity grade > 1.14, and 67.65% of the patients without bone metastases have an anellipticity grade < 1.14.

These observations are summarized in Table 2.

These proposed cut-off values are used for creating an algorithm to select breast cancers with a high risk of developing bone metastases.

## 4. Discussion

### 4.1. Previous Studies

In previous studies, Baak et al. [34,35] compared morphometric data on the survival rate in patients with breast carcinoma and established some morphometric criteria to differentiate the survivors and non-survivors over 6.5 years. In their study, they used only nuclear morphometry and compared the mean values and the standard deviations, regardless of tumor pattern and cell density in the tumor. They observed that survivors had a lower mitotic activity index (9.2 vs. 24.5) and lower mean nuclear perimeter (28.4 vs. 31.5), mean nuclear area (54.8 vs. 67.8), shorter smallest axis (7.2 vs. 8.1) and shorter longest axis (9.9 vs. 11).

In a more recent study, Kashyap et al. [36] studied FNAC specimens from breast cancer patients and showed that nuclear size parameters are increasing with cytological grades. Moreover, by setting as cut-off values 31.93 µm^2^ for mean nuclear area, 6.325 µm for equivalent diameter, 5.865 µm for minimum Feret, 7.855 µm for maximum Feret and 21.55 µm for mean nuclear perimeter, they were able to differentiate between benign and malignant specimens with high probability.

Our study comes out as a completion of the previous results given by different authors in order to obtain more results that are helpful for patient management.

### 4.2. Discussions on the Selected Morphometrical Parameters’ Values

The nuclear size parameters obtained in this study are comparable with the ones described in the above-mentioned sources, but this study also evaluated the dysmorphism of nuclei by using acyclicity grade and anellipticity grade, the repartition of cell components by using N/C ratio and the density of cells in the tumor by using mean internuclear distance. Moreover, the nuclei are compared at a higher level as, in this paper, not only the area is evaluated but also the long axis, the small axis and the nuclear volume, being considered an ellipsoid.

The cells from tumors with associated bone metastases seemed to be much smaller, as the nuclear area, the nuclear axes and the nuclear volume are smaller, and the N/C ratio is higher. This fact can be due to a higher probability of entering through the wall of small arteries and capillaries and also of entering between dense collagen fibers. The high collagenous environment can also be related to the smaller internuclear distance, as development between long and thick collagen fibers is harder due to mechanical factors. The high axis ratio and acyclicity grade can also explain the high probability of entering the bone environment, as ovoid deformed structures can more easily pass through the fenestrated capillaries. 

The acyclicity grade is able to evaluate the deviation from classical round form during the turn to malignant cells, while the anellipticity grade evaluates the dysmorphism of the elliptic nucleus. 

### 4.3. Elaboration of the Morphometrical Panel

Starting from the observations presented among results, there could be elaborated a morphometrical panel that can help with differentiating between the tumors associated with bone metastasis and the ones that do not associate. This morphometrical panel includes only eight criteria, as axis ratio is a derivative parameter from the other two considered parameters (both axes), and it was observed that it can be excluded without affecting the results.

Considering as criteria the eight parameters (without axes ratio) with their cut-off values for breast cancer patients with associated bone metastases, which are given in Table 2, it was observed that all the patients with breast cancer with associated bone metastases respected at least six out of eight criteria, while 94.11% of the patients without known bone metastases had at most five out of eight criteria respected. The considerations given above can be observed in Figure 3.

This way, it is believed that the breast cancer patients with at least six out of eight criteria are at high risk of developing bone metastasis, and they should be considered for more aggressive adjuvant treatment and/or novel therapeutic strategies. The rest of the patients were divided into two other categories—the ones with three, four or five criteria are considered at intermediate risk, and the ones with less than three criteria are included in the low-risk category.

### 4.4. The Novelty of the Method in the Context of Modern Pathology

This was the first time a morphometrical method was used for evaluating the risk of developing bone metastases. Most methods are based on molecular profile, morphological features of the tumor mass and other clinicopathological features of the breast cancer patients [37,38,39]. The proposed method has some advantages that will be mentioned below.

One of them regards the cost effectiveness as, once the apparatus is acquired, the slides scanning, the measurements and the results interpretation do not require supplementary costs.

Another advantage regards the time efficiency; the entire process requires a couple of hours and can be carried out in a pathology department without special conditions, as it does not use special reagents or special conditions for evaluation.

The third advantage is represented by the possibility of automatization. As the proposed algorithm is mathematical- and criteria-based, it can be easily integrated in a code such the Matlab R2022b one given in this paper and further integrated in more complex software.

### 4.5. Concept Limitations

The main disadvantage is represented by the current low accessibility to most pathology laboratories of whole slide imaging techniques, as well as the low number of physicians trained for morphometry.

The proposed algorithm requires special equipment (e.g., Olympus VS200 slide scanner) and software (e.g., QuPath v0.4.4.) in order to obtain the input data for the Matlab code. After obtaining the input data (mean nuclear area, mean nuclear perimeter, mean cell surface and mean length of the nuclear long axis), the entire algorithm can be calculated manually and complex specialized software such as Matlab is not mandatory. The idea of a Matlab code comes as an automatization future perspective in the era of artificial intelligence. Developing software would require code implementation.

This is an experimental study showing preliminary data that has not been validated yet. Our current focus is on extending our database of cases by collecting additional samples from tertiary centers in our region, in order to improve the accuracy of the algorithm.

### 4.6. Automatization in the Era of Digital Pathology and Artificial Intelligence

As the proposed algorithm is mathematically developed, it can be easily solved by coding. Therefore, the next Matlab code is proposed (Algorithm 1):
**Algorithm 1**  %in practice, there are two situations  %the first situation consists in the case in which the cells  %are individually measures by using Qupath  %and the second one consists in non-distinctive cell borders and  %the cell area is calculated as the ratio  %between the whole selected surface and the number of nuclei  %the data needed to be given
  %nuclear area mean value  A_nc=input(‘The mean value of nuclear area (µm^2) is’)  %nuclear mean perimeter  P_nc=input(‘The mean value of nuclear perimeter (µm) is’)  %cell mean surface value  S_cell=input(‘The mean value of cell surface (µm^2) is’)  %mean nuclear long axis length  L_nc=input(‘The mean length of nuclear long axis (µm) is’)  %as mentioned in results and discussions, there are 8 criteria  %that should be evaluated for a better result   %they are noted  %Crit1—mean nuclear surface (µm^2)  Crit1=A_nc;  %Crit2—surface ratio of nucleus to cytoplasm  A_cytoplasm=S_cell-A_nc;  Crit2=A_nc/A_cytoplasm;  %it is mentioned that this criterion can be calculated for each cell  %independently if they are measured individually and the mean value can  %be calculated as the mean value of all ratios  %Crit3—the aciclycity grade  %from the mean area of a nucleus, the mean radius can be calculated  r_nc=(A_nc/pi)^(1/2);  %therefore the perimeter of a circle with this radius is  P_nc_circle=2*pi*r_nc;  %the aciclycity grade is defined as the ratio between  %the perimeter of nucleus which is practically measured on slide  %and the perimeter of cycle with the same area as nucleus  Crit3=P_nc/P_nc_circle;  %Crit4—mean distance between nuclei  %in the hypotheses discussed, the mean distance can be approximated with  Crit4=S_cell^(1/2);  %Crit5—mean long axis length  %this parameter value is given as introductory value in this code  Crit5=L_nc;  %Crit6—mean small axis length  %in order to evaluate this criterion, the radius of the small axis  %has to be calculated in the hypothesis of an eliptic nucleus  %with the given mean area  %the half on long axis length is  R_nc=L_nc/2;  %from the formulae for elipse area  r_nc=A_nc/pi/R_nc;  %and now the small axis length can be calculated  Crit6=2*r_nc;  %Crit7—the mean nuclear volume  %for approximation of volume, the nucleus is considered an ellipsoid   %with the third axis length equal to the small axis length  %by knowing the formulae for an ellipsoid volume  Crit7=4/3*pi*R_nc*r_nc^2;  %Crit8—the anellipticity grade  %this last criterion is defined as the ratio between  %the nucleus perimeter and the perimeter of an ellipse with the same axes  %the perimeter of an ellipse is approximated with the given formulae  P_nc_ellipse=2*pi*((R_nc^2+r_nc^2)/2);  Crit8=P_nc/P_nc_ellipse;  %now that all the 8 criteria are defined,   %the selection of patients into risk category can begin  %a parameter for number of respected criteria has to be defined  n_cr=0;  %now each criterion should be evaluated with the proposed cut-off value  %evaluation of Crit1  if  Crit1<55  display(‘The first criterion is respected as the mean nuclear area is smaller than 55 µm^2’)  n_cr=n_cr+1;  else  display(‘The first criterion is not respected as the mean nuclear area is higher than 55 µm^2’)  end  %evaluation of Crit2  if Crit2>1.1  display(‘The second criterion is respected as the surface ratio of nucleus to cytoplasm is higher than 1.1’)  n_cr=n_cr+1;  else  display(‘The second criterion is not respected as the surface ratio of nucleus to cytoplasm is lower than 1.1’)  end  %evaluation of Crit3  if Crit3>1.145  display(‘The third criterion is respected as the acyclicity grade is higher than 1.145’)  n_cr=n_cr+1;  else  display(‘The third criterion is not respected as the acyclicity grade is smaller than 1.145’)  end  %evaluation of Crit4  if Crit4<10.5  display(‘The forth criterion is respected as the mean internuclear distance is smaller than 10.5 µm’)  n_cr=n_cr+1;  else  display(‘The forth criterion is not respected as the mean internuclear distance is higher than 10.5 µm’)  end  %evaluation of Crit5  if Crit5<9  display(‘The fifth criterion is respected as the mean long axis length is smaller than 9 µm’)  n_cr=n_cr+1;  else  display(‘The fifth criterion is not respected as the mean long axis length is larger than 9 µm’)  end  %evaluation of Crit6  if Crit6<7  display(‘The sixth criterion is respected as the mean small axis length is smaller than 7 µm’)  n_cr=n_cr+1;  else  display(‘The sixth criterion is not respected as the mean small axis length is larger than 7 µm’)  end  %evaluation of Crit7  if Crit7<200  display(‘The seventh criterion is respected as the mean nuclear volume is smaller than 200 µm^3’)  n_cr=n_cr+1;  else  display(‘The seventh criterion is not respected as the mean nuclear volume is larger than 200 µm^3’)  end  %evaluation of Crit8  if Crit8>1.14  display(‘The eight criterion is respected as the anellipticity grade is higher than 1.14’)  n_cr=n_cr+1;  else  display(‘The eight criterion is not respected as the anellipticity grade is smaller than 1.14’)  end  %all the eight criteria have been evaluated  %therefore the patient can be included in a category risk  if n_cr>5  display(‘The patient is at high risk category and the number of respected criteria is’)  n_cr  else   if  n_cr>2  display(‘The patient is at intermediate risk category and the number of respected criteria is’)  n_cr  else  display(‘The patient is at low risk category and the number of respected criteria is’)  n_cr  end  end

The proposed code facilitates the stratification of patients in a risk category by introducing simple measurement data such as mean nuclear area, cell area, mean long axis length and nuclear perimeter.

### 4.7. Summary of the Entire Process

The entire process can be summarized in the following steps:After establishing the diagnosis, a representative slide has to be selected. This slide should be chosen according to the criteria given in the methods section.The slide has to be scanned to create a virtual slide that can be analyzed by using specialized morphometric software such as QuPath.For the cells that meet the conditions given in the methods section, the nuclear area, the nuclear perimeter, the whole cell surface, and the long axis of the nucleus have to be measured.The eight criteria can be checked manually or by using the Matlab code, and the number of met criteria should be noted.Based on the number of criteria, the breast cancer patients can be assigned to a risk category.

## 5. Conclusions

The parameters evaluated for the characterization of breast cancers with associated bone metastases seem to be reliable for the evaluation of tumors and selecting the ones at risk of developing bone metastasis.

Using the established cut-off values, a novel algorithm is proposed, using specific calculation software, in order to stratify lesions with a high risk of developing bone metastases. All breast cancer patients with associated bone metastases had at least six out of eight criteria fulfilled. The patients with at least six criteria fulfilled are considered to be at high risk of developing bone metastases, while the ones with three, four or five criteria have been considered to have an intermediate risk, and the ones with less than three criteria are considered to be at low risk.

This algorithm aims to help in the selection of patients who should be further searched for bone metastasis or undergo molecular profile evaluation. Because it is mathematically obtained, it can be easily solved by using specific calculation software such as Matlab.

If the Matlab code is used, the number of criteria is automatically counted, and the risk category is automatically assigned. Therefore, the use of the Matlab code eliminates the need for mathematical calculations, criteria checking, manual counting of met criteria and manual assignment of patients to risk categories, as all these are performed automatically.

Although this method has limited application in current daily clinical practice, we strongly believe that further research and improvement of the proposed algorithm may eventually lead to the standardization of a more personalized therapeutic approach, especially with the advent of artificial intelligence and as digital pathology hopefully becomes readily available in more pathology labs around the world.

## Figures and Tables

**Figure 1 diagnostics-13-03338-f001:**
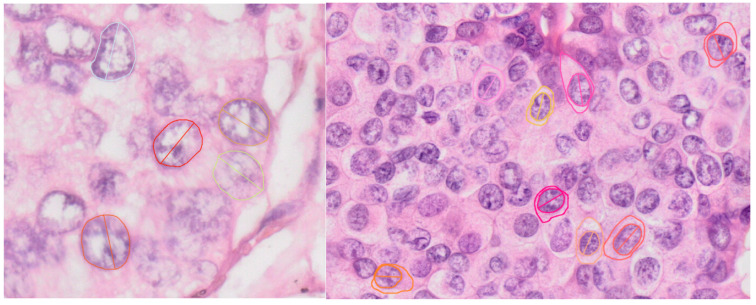
The measurements of nuclear dimensions and long axis. In first case (**left**), the cellular membrane cannot be identified on slides, so it was not measured, while, in the second case (**right**), the cellular membrane could be evaluated and measured.

**Figure 2 diagnostics-13-03338-f002:**
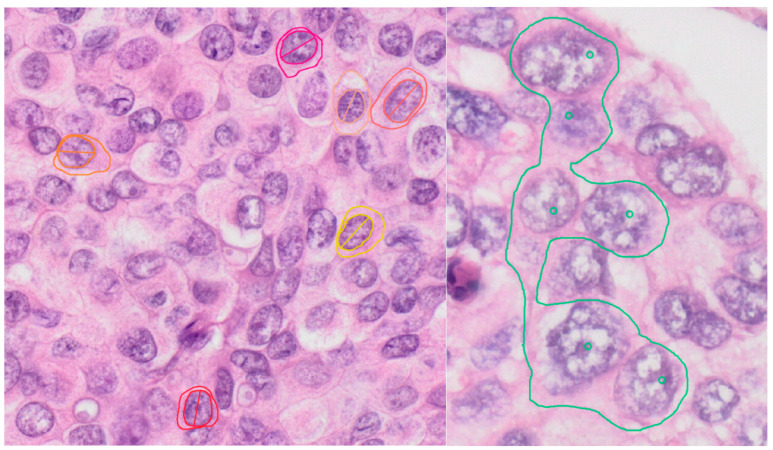
The evaluation and quantification of cytoplasm in tumor cells. In first case (**left**), the cellular membrane can be easily identified on slides, so it was not measured, while, in the second case (**right**), the cellular membrane could not be evaluated and measured, so areas including more cells were measured and the nuclei number was counted.

**Figure 3 diagnostics-13-03338-f003:**
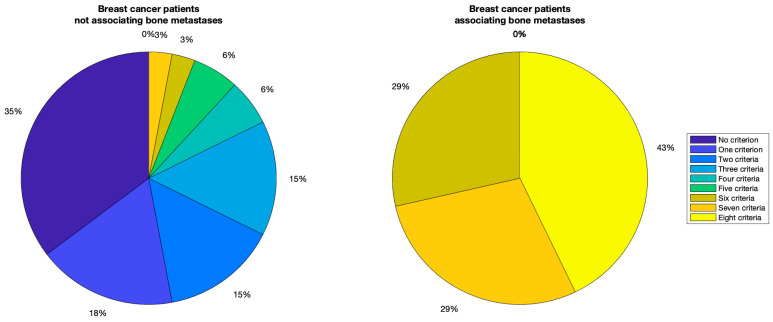
The comparison of breast cancer patients by number of respected criteria.

**Table 1 diagnostics-13-03338-t001:** Results for the considered parameters. * N/C represents the surface ratio of nucleus to cytoplasm (all the *p* values were calculated using regression analysis).

Evaluated Parameter	Breast Cancer with Associated Bone Metastasis	Breast Cancer without Associated Bone Metastasis	*p* Value
Average nuclear area (µm^2^)	42.583 ± 3.765	64.718 (54.5–86.37)	0.0035
N/C ratio *	1.405 ± 0.62	0.561 (0.44–0.97)	0.0418
Axis ratio	1.323 ± 0.123	1.268 ± 0.063	0.0880
Acyclicity grade	1.175 (1.15–1.19)	1.145 ± 0.031	0.0857
Mean internuclear distance (µm)	8.574 ± 0.643	12.871 ± 3.039	0.0007
Long axis length (µm)	8.411 ± 0.619	10.630 ± 1.676	0.0015
Small axis length (µm)	6.404 ± 0.326	8.115 (7.26–9.24)	0.0008
Mean nuclear volume (µm^3^)	181.023 ± 22.373	347.615 (262.15–525.69)	0.0146
Anellipticity grade	1.151 (1.14–1.156)	1.126 ± 0.028	0.1362

**Table 2 diagnostics-13-03338-t002:** Cut-off values for the considered parameters. * N/C represents the surface ratio of nucleus to cytoplasm.

Evaluated Parameter	Cut-Off Value	Breast Cancer Associating Bone Metastasis	Breast Cancer Not Associating Bone Metastasis
Average nuclear area (µm^2^)	55	<55	>55
N/C ratio *	1.1	>1.1	<1.1
Axis ratio	1.2	>1.2	<1.2
Acyclicity grade	1.145	>1.145	<1.145
Mean internuclear distance (µm)	10.5	<10.5	>10.5
Long axis length (µm)	9	<9	>9
Small axis length (µm)	7	<7	>7
Mean nuclear volume (µm^3^)	200	<200	>200
Anellipticity grade	1.14	>1.14	<1.14

## Data Availability

Data available on request due to restrictions, e.g., privacy or ethical.

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
