# Peer review of "A Novel Algorithm for Evaluating Bone Metastatic Potential of Breast Cancer through Morphometry and Computational Mathematics"

_diagnostics, 2023, doi:10.3390/diagnostics13213338_

Round 1

Reviewer 1 Report

Comments and Suggestions for Authors
  1. The abstract should be structured more clearly to provide a concise yet comprehensive overview of the paper. Consider breaking it down into sections like Background, Methodology, Results, and Conclusions.
  2. I have found some grammatical errors; kindly proofread the manuscript.
  3. While the abstract mentions findings such as "lower nuclear area" and "higher nucleus to cytoplasm ratio," it would be more informative if you could include specific numerical values or ranges to illustrate the magnitude of these differences.
  4. Provide a bit more context about the significance of bone metastases in breast cancer and why identifying high-risk patients is important. The author should add more details.
  5. This paper proposes a novel algorithm but lacks specific details about how it works. How would a researcher be able to understand your methodology clearly and provide a brief description of the algorithm's methodology or steps?
  6. Integrating some key findings from the results section into the conclusion will provide a more seamless transition and reinforcement of your main findings. or add a separate section for "our contributions."
  7. The paper should address the sample size and its representativeness. If the sample is small or not diverse, it could limit the generalizability of the algorithm.
  8. Does your algorithm have fault tolerance?
  9. The manuscript needs to be better structured, if possible, with clear headings and subheadings that guide the reader through the research process.
  10. All the references should be in ascending order, like after ref [14], ref 15 should come. But instead of ref. 15, ref. 17 comes. correct it.
  11. The paper lacks information about the validation of the proposed algorithm. The author suggested demonstrating the algorithm's reliability and accuracy using independent datasets or cross-validation.
  12. The paper mentions limited application in current daily clinical practice. Discuss the practical challenges and limitations that might hinder the immediate adoption of this algorithm in real clinical settings.
  13. Address the complexity of the algorithm and whether it requires specialized software (e.g., Matlab) for implementation.
  14. While the paper briefly mentions the potential for further research and improvement of the algorithm, it would be better to provide more specific insights into what aspects or challenges you plan to address in future studies.
  15. Discuss any ethical considerations related to patient data and the use of digital pathology in medical practice. Have you addressed data privacy and security concerns?
  16. I don't think the author has considered any external factors or variables (e.g., patient demographics, treatment history) that might impact the algorithm's accuracy and applicability.
  17. Compare the performance of your algorithm to existing methods or tools for evaluating bone metastatic potential (latest, 2023).

Comments on the Quality of English Language

require proof reading 

Reviewer 2 Report

Comments and Suggestions for Authors

The manuscript" A novel algorithm for evaluating bone metastatic potential of breast cancer through morphometry and computational mathematics" aims to identify an algorithm to evaluate breast metastsasis on the bone. The manuscript has clinical and scientific value. However, there are some comments to be addressed:

- The introduction is poorly structured. It is only based on breast cancer. The authors never mention the importance of digital pathology on breast cancer diagnosis on primary lesions or metastatic ones. There are several publications on the issue.

- Materials and Methods: Please provide the details about the digitalization. It was Whole-slide image?

- There is a concern on "representative area has been selected". It should not have been the entire lesion? What criteria were used to select the area?

- There are already in the literature other algorithms based in different characteristics than the ones you used. An algorithm just on those charateristics seems very elementar. Do you use Annotation or deconvolution of the images?

- Do you try the algorithm in different lesions? The histological parameters that you based the algorithm are not specific for breast cancer.

- The discussion should be improved based on other algorithms that are already published.

Comments on the Quality of English Language

Reviewer 3 Report

Comments and Suggestions for Authors

 Line 57 and line 63 apparently have contradictory data. Please explain.

Line 125 QuPath has no reference in the bibliography. Please add one.

What was the version of the QuPath? Please add.

Line 134 Why were chosen only 60 to 100 cells, when in WSI it´s available thousands of cells?

Table 1 – All variables should have the mean value and the standard deviation if normally distributed. If not, median and 25-75 percentil should be provided.

Line 231- Most cutoffs were chosen to be very specific to cancer with metastases and were chosen manually, which is always debatable. What was the purpose of choosing such cutoffs?

Line 260 – Table 2 is completely useless.

Line 265 – What kind of algorithm? Regression model? Other? Even to create an algorithm, one does not necessarily need to make cutoffs. Please explain.

Line 294 – How do you explain that HER2-positive and TNBC, that have higher nuclear grade, have more likely metastases than luminal A? The explanation doesn’t seem legit. Unless the author are saying that the low size of cancer nuclei are more likely to enter the bone tissue.

Line 327 – Matlab code? Where does this come from? Matlab is not open source and requires specific math knowledge. What is the main advantage to use Matlab compared to counting the number of morphometric features related with bone metastases stated in the study?

Comments on the Quality of English Language

Minor editing of English language required

Reviewer 4 Report

Comments and Suggestions for Authors

What are the other methods in the domain, how does this method compare with others.

How are the ranges for the measurements decided

Round 2

Reviewer 1 Report

Comments and Suggestions for Authors

Accept in current form

Author Response

Thank you very much!

Reviewer 3 Report

Comments and Suggestions for Authors

Reference 33 - QuPath request this paper to be cited when using the software: Bankhead, P. et al. QuPath: Open source software for digital pathology image analysisScientific Reports (2017).

Table 1 - p value needs an explanaition of wich test was performed in each feature

Comments on the Quality of English Language

Minor editing of English language required

Reviewer 4 Report

Comments and Suggestions for Authors

The authors have answered all the queries.

Author Response

Thank you very much!